# Diagnostic Method for Short Circuit Faults at the Generator End of Ship Power Systems Based on MWDN and Deep-Gated RNN-FCN

**Lanyong Zhang** [1,2,*], **Ziqi Zhang** [1] and **Huimin Peng** [2]

1   College of Intelligent Systems Science and Engineering, Harbin Engineering University, Harbin 150001, China; zhangziqi@hrbeu.edu.cn
2   Nari Group Corporation, Nanjing 211106, China; penghuimin@sgepri.sgcc.com.cn
*   Correspondence: zhanglanyong@hrbeu.edu.cn

**Abstract:** Synchronous generators with three phases are crucial components of modern integrated power systems in ships. These generators provide power for the entire operation of the vessel. Therefore, it is of paramount importance to diagnose short-circuit faults at the generator terminal in the ship's power system to ensure the safe and stable operation of modern ships. In this study, a generator terminal short-circuit fault diagnosis method is proposed based on a hybrid model that combines the Multi-Level Wavelet Decomposition Network, Deep-Gated Recurrent Neural Network, and Fully Convolutional Network. Firstly, the Multi-Level Wavelet Decomposition Network is used to decompose and denoise the collected electrical signals, thus dividing them into sub-signals and extracting their time-domain and frequency-domain features. Secondly, synthetic oversampling based on Gaussian random variables is employed to address the problem of imbalance between normal data and fault data, resulting in a balanced dataset. Finally, the dataset is fed into the hybrid model of the Deep-Gated Recurrent Neural Network and Fully Convolutional Network for feature extraction and classification of faults, ultimately outputting the fault diagnosis results. To validate the performance of the proposed method, simulations and comparative analysis with other algorithms are conducted on the fault diagnosis method. The proposed algorithm's accuracy reaches 96.82%, precision reaches 97.35%, and the area under curve reaches 0.85, indicating accurate feature extraction and classification for identifying short-circuit faults at the generator terminals.

**Keywords:** ship integrated power system; fault diagnosis; multi-level wavelet decomposition network; improved SMOTE algorithm; deep-gated RNN-FCN

## 1. Introduction

Fully electrically propelled ships are becoming increasingly popular due to their economic efficiency and high reliability, making them the primary focus for the development of advanced ships. However, these ships can have high power demands of up to 100 MW [1]. The integrated power propulsion system of a ship can not only meet this critical technological requirement but can also provide power to various onboard loads, making it a crucial trend for the development of ship power systems. Currently, the power systems of fully electrically propelled ships are primarily classified into three types: medium-voltage AC, medium-voltage DC, and high-frequency AC, with medium-voltage AC and high-frequency AC power systems being the most widely used. However, these systems have some inherent drawbacks, such as difficulties in grid connection and poor performance in motor speed control [2]. Therefore, researchers are focusing on MVDC power systems as a way to overcome these challenges.

Modern ship's integrated power systems require high-capacity three-phase synchronous generator sets, which typically include permanent magnet synchronous generators, induction synchronous generators, slip-ring synchronous generators, and air-gap

synchronous generators [3]. These generators provide stable three-phase voltage and frequency output for ships, and there are usually multiple generators connected to different electrical equipment and systems. During operation, shipboard synchronous generators store a significant amount of electromagnetic energy internally and continue to rotate even in the event of an external short-circuit fault due to the physical characteristics of the generator. As a result, the conversion time of electromagnetic energy is relatively long [4]. Short-circuit faults are the most common failures in modern integrated electric propulsion systems for ships and can have a significant impact, causing abnormal increases in current, voltage fluctuations, and even fire hazards [5]. To ensure the safety and reliable operation of the ship, accurate diagnosis of short-circuit faults at the output terminal of synchronous generators, including timely diagnosis, detection, and repair, is necessary.

In the field of short-circuit fault diagnosis in ship's integrated power propulsion systems, the commonly used research methods can be classified into three categories: analytical model-based methods [6], data-driven methods [7], and qualitative model-based methods [8]. The specific classification is shown in Table 1 [9]. The methods described above are widely used in the field of fault diagnosis in the ship's integrated power propulsion systems; however, each of them has its limitations. Analytical model-based fault diagnosis methods aim to diagnose faults in real time based on the essential characteristics of the object system. However, they require highly accurate mathematical models, which can be challenging to develop precisely due to the harsh operating environment and complex and varying operating conditions of shipboard power systems [10]. Data-driven methods require different processing approaches for different faults or signals, which limits their generality [11]. In contrast, qualitative model-based methods face difficulties when diagnosing complex systems, such as the ship's power propulsion systems. The number of fault branches rapidly increases once a fault occurs, making the diagnosis process increasingly complicated and resulting in lower accuracy and poor real-time performance of the diagnosis results [12].

**Table 1.** Common methods of fault diagnosis.

| | | |
|---|---|---|
| Failure diagnosis method | Method based on analytical model | State estimation method |
| | | Parameter estimation method |
| | | Equivalent spatial method |
| | Method based on a data-driven approach | Method based on signal processing — WT |
| | | EMD |
| | | MSP |
| | | Spectral Analysis |
| | | Multivariate statistical analysis — PCA |
| | | Method based on shallow machine learning — ANN |
| | | SVM |
| | | Method based on deep learning |
| | | Method based on information fusion — Fuzzy fusion method |
| | | Reliability function theory method |
| | | Rule reasoning |
| | Method based on qualitative model-based approach | Fault Tree Analysis |
| | | SDG |

With the emergence of AI technology and its subfields, including machine learning and deep learning, AI-based intelligent fault diagnosis methods have become a prominent trend in the field [13]. Typically, the main steps involved in these methods are data preprocessing, fault feature extraction, and fault classification. Unlike other methods, AI-based approaches do not rely on constructing accurate mathematical models for the research object, and they provide benefits, such as handling large datasets, efficient learning, high diagnostic accuracy, and versatility in various domains [14]. Figure 1 illustrates the general workflow involved in traditional AI-based intelligent fault diagnosis methods.

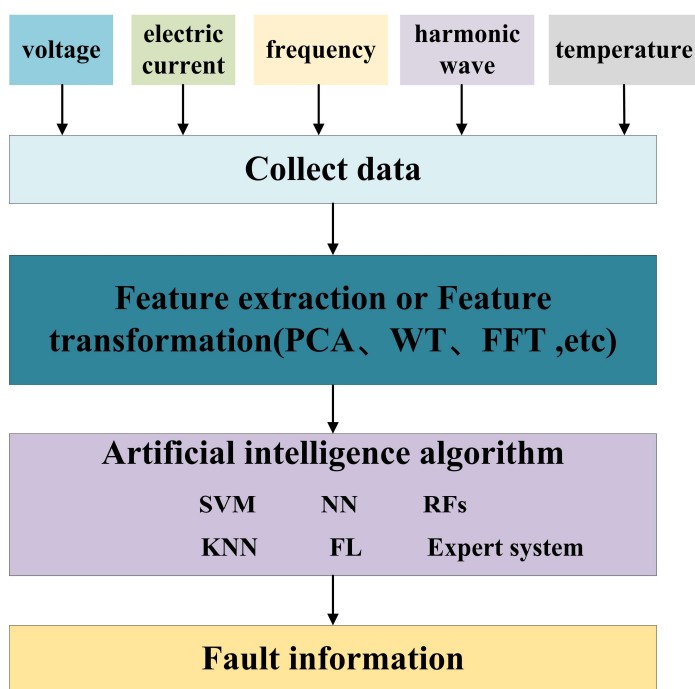

**Figure 1.** Workflow of AI-based intelligent fault diagnosis method.

Table 2 classifies traditional intelligent fault diagnosis methods [15]. Despite their utility, traditional approaches have four primary drawbacks. Firstly, they heavily rely on domain-specific expert experience and prior knowledge, resulting in limited generalization and suboptimal performance when detecting faults in different objects. Secondly, the harsh operating conditions of ship power systems generate significant amounts of noise, preventing traditional algorithms from extracting reliable features for detecting faults in generators [16]. Thirdly, ship power grid data exhibit the "4V" characteristics, leading to imbalanced datasets due to the generation of massive amounts of data during operation, coupled with limited fault data. Finally, power grid data are provided in time series format, which is not suitable for traditional algorithms to perform feature extraction and classification. Consequently, AI-based intelligent fault diagnosis methods have become a topic of significant interest for research in addressing these challenges [17]. Chaochun Yu et al. [18] have proposed an improved CNN-based network fault diagnosis model for detecting faults in synchronous generators of ship power systems. This method eliminates the requirement for feature decomposition and extraction, making it simple to implement and able to operate at high speeds. Nonetheless, the input dataset for this method comprises simulated fault images, and preparing a large-scale image dataset entails a laborious and time-consuming process for collecting and labeling the data.

**Table 2.** Intelligent fault diagnosis methods based on artificial intelligence.

| | | |
|---|---|---|
| Fault diagnosis method based on artificial intelligence | Supervised learning | Artificial Neural Network |
| | | Support Vector Machine |
| | | Random Forests |
| | | K-nearest neighbor |
| | Semi-supervised learning | Generative method |
| | | A divergence-based approach |
| | Unsupervised learning | Self Organizing Map |
| | | Clustering algorithm |
| | | Principal Component Analysis |
| | Deep learning | Convolutional Neural Network |
| | | Deep Belief Network |
| | | Stacked Auto Encoders |
| | Reinforcement learning | Q-learning |
| | Transfer learning | TrAda Boost |

Wenxin Liu et al. [19] tackled the issue of diagnosing stator winding faults in permanent magnet synchronous motors by proposing an improved PSO algorithm to optimize the difficult parameter selection. However, the dependence on global optimization limits the PSO algorithm's ability to model high-dimensional data sequences.

Mengshi Li et al. [20] proposed a data-driven technique for wind turbine fault diagnosis based on the attributes of power grid data sequences. This method implemented an LSTM network, applying residual generators and autonomous decision-making based on the Architecture RF. The LSTM network showed excellent results in processing time series data of wind turbine units. However, due to the computation requirements of using a single LSTM for long sequences, training and inference speeds are comparatively slower.

After considering the advantages and disadvantages of traditional artificial intelligence algorithms, this paper proposes an innovative fault diagnosis method for short circuit faults in the three-phase synchronous generator output of a ship's MVDC power propulsion system, utilizing MWDN and the fusion of deep-gated RNN-FCN. Firstly, MWDN is employed to preprocess the power grid data by performing signal decomposition and denoising to facilitate further feature extraction. Next, the issue of imbalanced sample data is addressed by using SMOTE, improved by Gaussian random variables, for oversampling the fault class samples and generating synthetic samples to balance the dataset. Finally, the deep-gated RNN-FCN network is utilized to extract features and classify faults from the preprocessed dataset. This is performed by introducing the dropout regularization technique after each deep-gated RNN submodule to reduce the risk of overfitting. Experiments show that the developed method achieves high accuracy, thus presenting innovative contributions to fault diagnosis, including the following:

(1) A novel data preprocessing method based on MWDN is employed to extract both time-domain and frequency-domain features of the power grid signals. The method decomposes the collected electrical signals into sub-signals, adds the denoising process, and improves the accuracy of fault diagnosis by reducing the impact of noise on feature extraction from original signals.

(2) To address the issue of imbalanced sample data, a SMOTE algorithm based on Gaussian random variables is proposed. While the SMOTE algorithm can handle the class imbalance problem in power grid data, the artificially synthesized new samples tend to lie on the same straight line, resulting in an issue of overgeneralization. By introducing Gaussian probability distribution into the feature space, the SMOTE algo-

rithm based on Gaussian random variables is developed, which allows the artificially generated samples to deviate from the straight line, thereby addressing this problem.

(3) A fault diagnosis method based on deep-gated RNN-FCN is introduced, with hidden state calculation performed by LSTM, creating a deep-gated RNN. Additionally, a deep-gated RNN submodule is added to the FCN module, expanding the FCN into a deep-gated RNN-FCN and improving fault feature extraction. Softmax function is utilized in the classification layer to address the problem of gradient vanishing and capture long-distance dependencies in the time series data, leading to enhanced network performance.

This paper is structured as follows. In Section 2, the structure of the synchronous generator is explained, including its mathematical models for voltage, magnetic flux, and short-circuit current, and fault labels are provided. Section 3 presents the overall framework of the proposed fault diagnosis algorithm, along with detailed descriptions of each component. In Section 4, simulations of the proposed algorithm are conducted and compared with other algorithms. Based on the simulation results, a comprehensive evaluation of the performance of the proposed method is provided. Finally, Section 5 concludes the paper.

## 2. Mathematical Model and Fault Analysis of Synchronous Generators

### 2.1. Mathematical Modeling of Voltage, Magnetic Flux, and Short-Circuit Current in Synchronous Generators

When a short-circuit fault happens at the synchronous generator's terminal, mathematical models can be utilized for fault diagnosis, analysis, and prediction. By incorporating effects such as changes in current, voltage, and power resulting from the fault, as well as fault propagation and diffusion mechanisms, mathematical models offer valuable insights for analyzing faults. Additionally, mathematical models can aid in fault diagnosis and localization.

Synchronous generators can be classified into two types based on their structure: rotating armature synchronous generators and rotating field synchronous generators. Currently, rotating field synchronous generators are widely used in medium to large-sized motors. The two types of rotating field synchronous generators are salient pole synchronous generators and non-salient pole synchronous generators, as illustrated in Figure 2, which depict their basic structures.

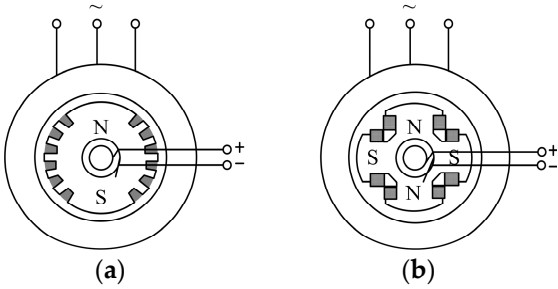

(a)  (b)

**Figure 2.** Two basic forms of rotating field synchronous generators: (**a**) salient pole and (**b**) non-salient pole.

This study focuses on the investigation of salient pole three-phase synchronous generators. The dynamic characteristics of synchronous generators are complex, with intricate coupling relationships existing among different windings. Therefore, modeling the generator using a three-phase stationary coordinate system would result in a significant computational burden. As a solution, this paper utilizes a dq rotating coordinate system to transform the three-phase coordinate system, enabling decoupling between the different magnetic fluxes in the generator. This reduces the number of variables in the generator

equations and improves computational efficiency. The following is the Park transformation matrix for dq0 coordinate conversion:

$$P = \frac{2}{3} \begin{bmatrix} \cos\theta & \cos(\theta - \frac{2\pi}{3}) & \cos(\theta + \frac{2\pi}{3}) \\ \sin\theta & -\sin(\theta - \frac{2\pi}{3}) & -\sin(\theta + \frac{2\pi}{3}) \\ \frac{1}{2} & \frac{1}{2} & \frac{1}{2} \end{bmatrix} \tag{1}$$

The transformed flux equations are as follows:

$$\begin{bmatrix} \varphi_d \\ \varphi_q \\ \varphi_f \\ \varphi_D \\ \varphi_Q \end{bmatrix} = \begin{bmatrix} X_d & 0 & X_{ad} & X_{ad} & 0 \\ 0 & X_q & 0 & 0 & X_{aq} \\ X_{ad} & 0 & X_f & X_{ad} & 0 \\ X_{ad} & 0 & X_{ad} & X_D & 0 \\ 0 & X_{aq} & 0 & 0 & X_Q \end{bmatrix} \begin{bmatrix} -i_d \\ -i_q \\ i_f \\ i_D \\ i_Q \end{bmatrix} \tag{2}$$

The transformed voltage equations are as follows:

$$\begin{bmatrix} u_d \\ u_q \\ u_f \\ u_D \\ u_Q \end{bmatrix} = \frac{d}{dt} \begin{bmatrix} \varphi_d \\ \varphi_q \\ \varphi_f \\ \varphi_D \\ \varphi_Q \end{bmatrix} + \begin{bmatrix} -r_a i_d \\ -r_a i_q \\ r_f i_f \\ r_D i_D \\ r_Q i_Q \end{bmatrix} + \begin{bmatrix} -\omega\varphi_q \\ \omega\varphi_d \\ 0 \\ 0 \\ 0 \end{bmatrix} \tag{3}$$

where $u_D = 0$ and $u_Q = 0$. The equation for the electromagnetic torque of the synchronous generator after the transformation is as follows:

$$M_e = \varphi_d i_q - \varphi_q i_d \tag{4}$$

In ship power systems, a short circuit fault can cause a transient process in the generator, resulting in a deviation of the terminal voltage and frequency from their steady-state values. Despite the short duration of the short circuit effect due to the large rotor inertia, it is still necessary to consider the electromagnetic transient process of the generator. Therefore, it is commonly assumed that the generator rotor maintains synchronous speed, keeping the frequency constant. However, it is still necessary to consider the electromagnetic transient process of the generator, which means that the terminal voltage cannot remain stable [21].

During normal operation, the three-phase stator currents of a synchronous generator are symmetrical and balanced positive-sequence currents, expressed by the following equation:

$$\begin{cases} i_a = I_m \cos(\omega t + \alpha_0) \\ i_b = I_m \cos(\omega t + \alpha_0 - \frac{2}{3}\pi) \\ i_c = I_m \cos(\omega t + \alpha_0 + \frac{2}{3}\pi) \end{cases} \tag{5}$$

When a short circuit occurs at a synchronous generator's terminal, the resulting fault current is equal to the sum of its AC and DC components [22]. The expression for the total stator three-phase current is given by

$$i_a(t) = \left[ \left( \frac{E''_{q0}}{X''_d} - \frac{E'_{q0}}{X'_d} \right) e^{-\frac{t}{T''_d}} + \left( \frac{E'_{q0}}{X'_d} - \frac{E_{q(0)}}{X_d} \right) e^{-\frac{t}{T'_d}} + \frac{E_{q(0)}}{X_d} \right] \cos(\omega t + \theta_0)$$
$$+ \frac{E''_{d0}}{X''_q} e^{-\frac{t}{T''_q}} \sin(\omega t + \theta_0)$$
$$- \frac{u_0}{2} \left( \frac{1}{X''_d} + \frac{1}{X''_q} \right) e^{-\frac{1}{T_a}} \cos(\delta_0 - \theta_0) - \frac{u_0}{2} \left( \frac{1}{X''_d} - \frac{1}{X''_q} \right) e^{-\frac{1}{T_a}} \cos(2\omega t + \delta_0 + \theta_0) \tag{6}$$

$$i_b(t) = [(\frac{E''_{q0}}{X''_d} - \frac{E'_{q0}}{X'_d})e^{-\frac{t}{T''_d}} + (\frac{E'_{q0}}{X'_d} - \frac{E_{q(0)}}{X_d})e^{-\frac{t}{T'_d}} + \frac{E_{q(0)}}{X_d}]\cos(\omega t + \theta_0 - \frac{2}{3}\pi)$$
$$+ \frac{E''_{d0}}{X''_q}e^{-\frac{t}{T''_q}}\sin(\omega t + \theta_0 - \frac{2}{3}\pi) \quad (7)$$
$$- \frac{u_0}{2}(\frac{1}{X''_d} + \frac{1}{X''_q})e^{-\frac{1}{T_a}}\cos(\delta_0 - \theta_0 - \frac{2}{3}\pi) - \frac{u_0}{2}(\frac{1}{X''_d} - \frac{1}{X''_q})e^{-\frac{1}{T_a}}\cos(2\omega t + \delta_0 + \theta_0 - \frac{2}{3}\pi)$$

$$i_c(t) = [(\frac{E''_{q0}}{X''_d} - \frac{E'_{q0}}{X'_d})e^{-\frac{t}{T''_d}} + (\frac{E'_{q0}}{X'_d} - \frac{E_{q(0)}}{X_d})e^{-\frac{t}{T'_d}} + \frac{E_{q(0)}}{X_d}]\cos(\omega t + \theta_0 + \frac{2}{3}\pi)$$
$$+ \frac{E''_{d0}}{X''_q}e^{-\frac{t}{T''_q}}\sin(\omega t + \theta_0 + \frac{2}{3}\pi) \quad (8)$$
$$- \frac{u_0}{2}(\frac{1}{X''_d} + \frac{1}{X''_q})e^{-\frac{1}{T_a}}\cos(\delta_0 - \theta_0 + \frac{2}{3}\pi) - \frac{u_0}{2}(\frac{1}{X''_d} - \frac{1}{X''_q})e^{-\frac{1}{T_a}}\cos(2\omega t + \delta_0 + \theta_0 + \frac{2}{3}\pi)$$

### 2.2. Fault Types and Labels

The occurrence of generator-end short circuit faults in ship power systems denotes a situation where one or several phases of the generator output terminal are short-circuited to the ground or between different phases. Such faults cause severe and common damage to the ship's power system during normal operation. Short circuit faults in ship power systems are classified into single-phase, two-phase, and three-phase depending on their specifics. These faults result in 11 fault state labels, which are presented in Table 3. The labeled dataset is used for training and testing the algorithm proposed in this paper.

**Table 3.** Correspondence between fault types and labels.

| Mode | Type | Label |
|---|---|---|
| Fault 1 | A-phase grounding | 1 |
| Fault 2 | B-phase grounding | 2 |
| Fault 3 | C-phase grounding | 3 |
| Fault 4 | A and B-phase short circuit grounding | 4 |
| Fault 5 | A and C-phase short circuit grounding | 5 |
| Fault 6 | B and C-phase short circuit grounding | 6 |
| Fault 7 | Three-phase short circuit grounding (A, B, and C phases) | 7 |
| Fault 8 | A and B-phase short circuit | 8 |
| Fault 9 | A and C-phase short circuit | 9 |
| Fault 10 | B and C-phase short circuit | 10 |
| Fault 11 | A, B, C three-phase short circuit | 11 |

## 3. Overall Framework and Detailed Explanation of Fault Diagnosis Algorithm

This paper proposes a novel fault diagnosis method based on MWDN and Deep-Gated RNN-FCN to diagnose the three-phase synchronous generator terminal short-circuit fault prevalent in ship MVDC power propulsion systems. The overall framework of the proposed method is illustrated in Figure 3. The model takes raw voltage and current data as input and uses MWDN to extract time-domain and frequency-domain features of the power system signals. The method addresses the class imbalance issue via SMOTE based on Gaussian random variables, generating artificial samples to balance the dataset. To achieve fault classification, the deep-gated RNN-FCN model is employed for feature extraction, with the output being the result of fault classification.

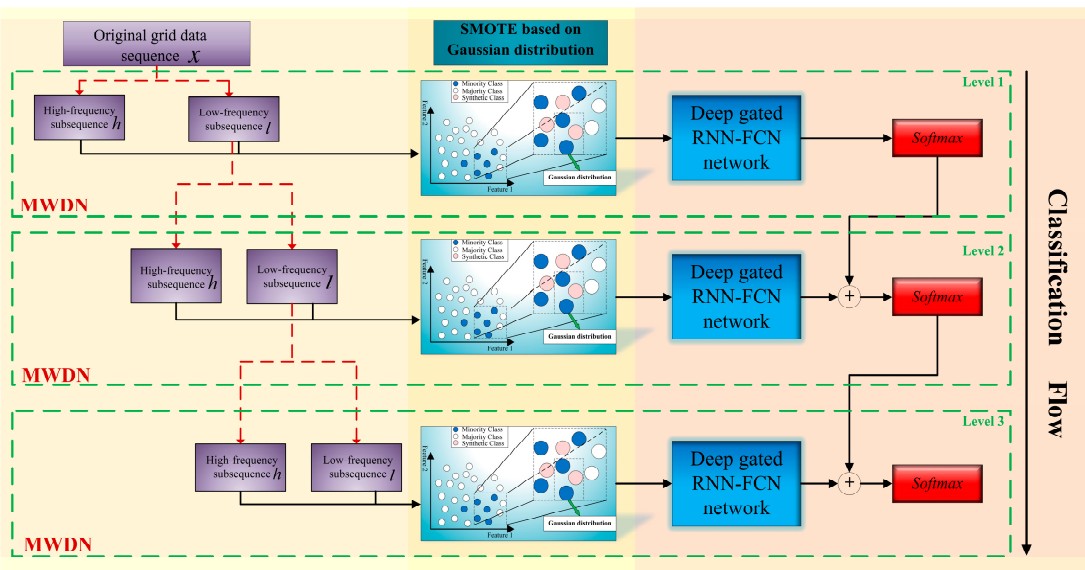

**Figure 3.** Overall framework of the fault diagnosis algorithm in this paper.

*3.1. Multi-Level Wavelet Decomposition Network*

MWDN can break down the fault data sequence in a power grid into a collection of high- to low-frequency sub-sequences. The MWDN method is an approximate MDWD that operates within the framework of a deep neural network, utilizing the temporal decomposition capability of wavelets while benefiting from the learning ability of deep neural networks. Compared to FFT, the MWDN technique excels in handling non-stationary signals, offering excellent time-frequency dual resolution and multi-scale analysis. It is especially useful for processing power grid fault signals as it displays energy concentration [23,24]. MWDN is similar in principle to MDWD recommended in reference [25]. Unlike MDWD, MWDN enables model parameter fine-tuning, enhancing its adaptability to data from various sources. Figure 4 shows the schematic diagram of the MWDN.

First, the principle of multilevel discrete wavelet decomposition is introduced. Let $x = \{x_1, \dots, x_t, \dots, x_T\}$ represent the input fault data sequence, $x_h(i)$ represent the high-frequency subsequence of the $i$-th level, $x_l(i)$ represent the low-frequency subsequence, $l$ represent the low-pass filter, $l = \{l_1, \dots, l_k, \dots, l_K\}$, $h$ represent the high-pass filter, $h = \{h_1, \dots, h_k, \dots, h_K\}$, where $K \ll T$. In the $i + 1$-th level, MDWD uses both the high-pass and low-pass filters to convolve the lower-level low-frequency subsequence, as shown in Equation (9):

$$
\begin{aligned}
a_n^l(i+1) &= \sum_{k=1}^{K} x_{n+k-1}^l(i) \bullet l_k \\
a_n^h(i+1) &= \sum_{k=1}^{K} x_{n+k-1}^l(i) \bullet h_k
\end{aligned}
\tag{9}
$$

The low-frequency subsequence $x^l(i)$ and the high-frequency subsequence $x^h(i)$ in the $i$-th level are generated by downsampling $a^l(i) = \left\{ a_1^l(i), a_2^l(i), a_3^l(i) \dots \right\}$ and $a^h(i) = \left\{ a_1^h(i), a_2^h(i), a_3^h(i) \dots \right\}$ by a factor of 1/2. $a^l(i) = \left\{ a_1^l(i), a_2^l(i), a_3^l(i) \dots \right\}$ and $a^h(i) = \left\{ a_1^h(i), a_2^h(i), a_3^h(i) \dots \right\}$ are intermediate variable sequences. $x^h(1), x^h(2), x^h(3), \dots, x^h(i)$, $x^l(i)$ together form the subset $X(i) = \left\{ x^h(1), x^h(2), x^h(3), \dots, x^h(i), x^l(i) \right\}$, which represents the $i$-th level decomposition result of the sequence $x$. Additionally, sequence $X(i)$ satisfies the following three conditions:

(1)  $x$ can be completely reconstructed from $X(i)$;
(2)  The frequency decreases sequentially from $x^h(1)$ to $x^l(i)$; i.e., it goes from high-frequency to low-frequency;

(3) At different levels, the time and frequency resolutions of the subset $X(i)$ are different. Moreover, as $i$ increases, the frequency resolution also increases, while the time resolution decreases. These phenomena are particularly evident in the low-frequency subsequence.

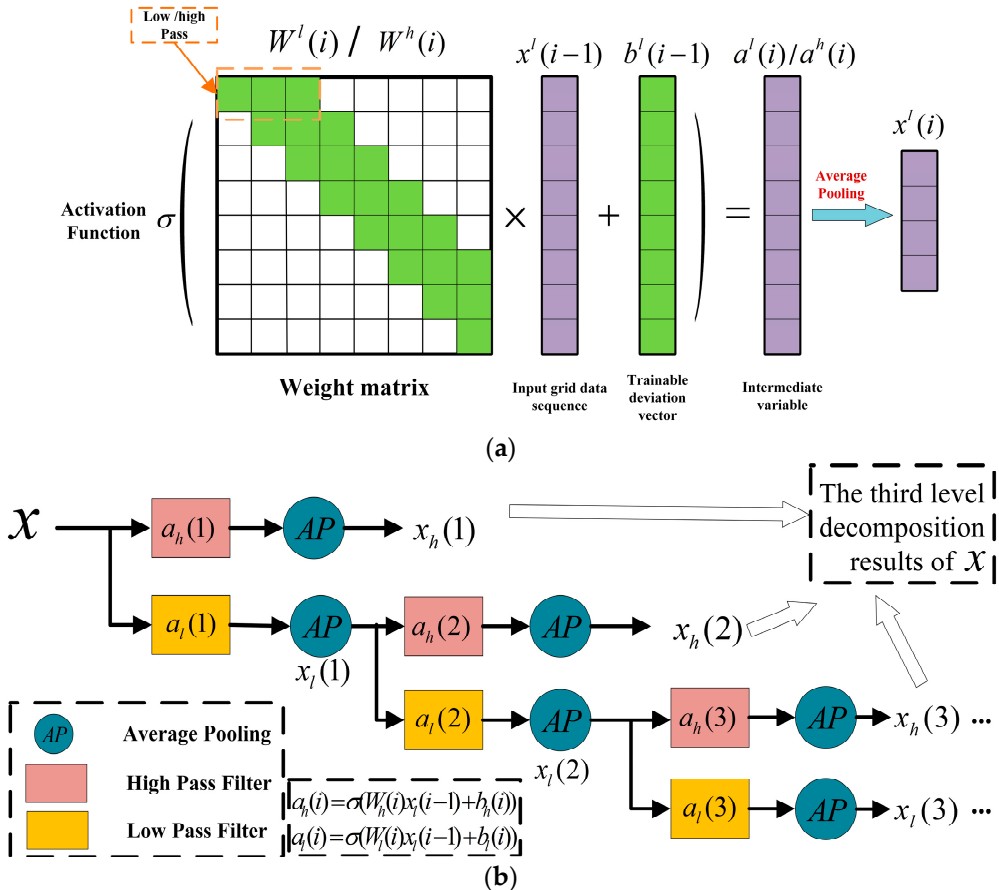

**Figure 4.** Schematic diagram of the multi-level wavelet decomposition network: (**a**) approximate MDWD and (**b**) MWDN framework.

Based on the principle of MDWD, we can obtain the framework of MWDN. As shown in Figure 4a, MWDN performs hierarchical decomposition of fault data sequences using functions $a^h(i)$ and $a^l(i)$:

$$a^h(i) = \sigma(W^h(i)x^l(i-1) + b^h(i))$$
$$a^l(i) = \sigma(W^l(i)x^l(i-1) + b^l(i))$$
(10)

The *sigmoid* function is used in MWDN. $x^h(i)$ and $x^l(i)$ have the same meanings as in MDWD, and $x_j^l(i) = \frac{(a_{2j}^l(i) + a_{2j-1}^l(i))}{2}$. The weight matrices $\mathbf{W^h}$ and $\mathbf{W^l}$ in Figure 4a are

trainable and belong to $R^{M \times M}$. $M$ represents the size of $x^l(i-1)$, $\mathbf{W^h}$ and $\mathbf{W^l}$ are defined as follows:

$$
\mathbf{W^h}(i) = \begin{bmatrix}
h_1 & h_2 & h_3 & \cdots & h_K & \in & \cdots & \in \\
\in & h_1 & h_2 & \cdots & h_{K-1} & h_K & \cdots & \in \\
\vdots & \vdots & \vdots & \ddots & \vdots & \vdots & \vdots & \vdots \\
\in & \in & \in & \cdots & h_1 & \cdots & h_{K-1} & h_K \\
\vdots & \vdots & \vdots & \ddots & \vdots & \vdots & \vdots & \vdots \\
\in & \in & \in & \cdots & \cdots & \cdots & h_1 & h_2 \\
\in & \in & \in & \cdots & \cdots & \cdots & \in & h_1
\end{bmatrix}
$$

$$
\mathbf{W^l}(i) = \begin{bmatrix}
l_1 & l_2 & l_3 & \cdots & l_K & \in & \cdots & \in \\
\in & l_1 & l_2 & \cdots & l_{K-1} & l_K & \cdots & \in \\
\vdots & \vdots & \vdots & \ddots & \vdots & \vdots & \vdots & \vdots \\
\in & \in & \in & \cdots & l_1 & \cdots & l_{K-1} & l_K \\
\vdots & \vdots & \vdots & \ddots & \vdots & \vdots & \vdots & \vdots \\
\in & \in & \in & \cdots & \cdots & \cdots & l_1 & l_2 \\
\in & \in & \in & \cdots & \cdots & \cdots & \in & l_1
\end{bmatrix}
\tag{11}
$$

where $|\in| \ll |l|, \forall l \in 1, |\in| \ll |h|, \forall h \in \mathbf{h}$. With this, the construction of MDWN is complete.

In this paper, MWDN is employed to decompose the voltage and current data collected during fault occurrences into high-frequency and low-frequency sub-sequences. This approach extracts frequency-domain and time-domain characteristics of the signals and applies denoising techniques to collected power system signals. The goal of this approach is to improve the final fault diagnosis accuracy.

*3.2. SMOTE Method Based on Gaussian Random Variable*

The data collected from ship power systems are characterized by a significant imbalance between the normal and faulty classes, with a large majority of normal operating data. The data imbalance presents challenges for fault diagnosis. This paper proposes an enhanced SMOTE optimization algorithm based on Gaussian random variables to address the class imbalance between the fault and normal classes.

The SMOTE [26] algorithm uses the KNN to obtain a uniform probability distribution and oversamples the fault class by generating synthetic samples [27]. First, the power system data is divided into normal and fault data. Next, the K-nearest neighbors are obtained using the KNN algorithm for each fault data point. A random selection of neighbors is then made within the K nearest neighbors for each fault sample to create synthetic new samples. Finally, the difference is computed between the fault sample data and its nearest neighbors, as depicted in Equation (12).

$$
dif = \left| C_{fault} - C_{NN}^k \right| \tag{12}
$$

The value $dif$, obtained from Equation (12), is multiplied by a random value following a uniform probability distribution to introduce randomness. Synthetic new samples can be generated using the following equation:

$$
C_{synthetic} = C_{fault} + \left| C_{fault} - C_{NN}^k \right| \times P_{random} \tag{13}
$$

This process is repeated until the desired number of synthetic samples is obtained.

The process described above refers to the standard SMOTE algorithm. Nonetheless, the synthetic samples generated by the standard SMOTE algorithm tend to lie on the same line, resulting in overfitting. To overcome this issue, we propose a SMOTE algorithm based on the Gaussian random variable. The calculation of the difference value in the SMOTE algorithm based on Gaussian random variables utilizes the same approach as

SMOTE. However, the improved algorithm makes a random selection between 0 and using Equation (14):

$$gap \sim U(0, dif) \tag{14}$$

With this equation, we can estimate potential locations for synthetic samples. Next, we follow Equation (15) and sample another number from a Gaussian distribution to use as the parameter $\sigma$:

$$range \sim N(gap, \sigma) \tag{15}$$

Equations (14) and (15) help derive the parameters required for generating synthetic data in the feature space, as depicted in Equation (16):

$$C_{synthetic} = C_{fault} + dif \times range \tag{16}$$

The SMOTE algorithm based on Gaussian random variables improves synthetic sample generation by utilizing parameters selected from a Gaussian distribution. As a result, synthetic samples can be created in locations beyond the straight line that connects fault class samples. Nonetheless, the algorithm guarantees that the synthetic samples remain near the line to ensure accuracy and reasonableness. Figure 5 shows the schematic diagram and flowchart of the algorithm.

In synthetic samples, overgeneralization is a persistent problem due to the significant class imbalance that necessitates the creation of synthetic data. Consequently, there is a high probability that the synthetic data will be located along the same line. This phenomenon represents a type of overgeneralization. To address this issue, this paper proposes the Gaussian random variable-based SMOTE method, which incorporates Gaussian probability distributions in the feature space. This approach permits the SMOTE algorithm's newly generated synthetic samples to deviate minimally from the dominant line due to the Gaussian probability distribution.

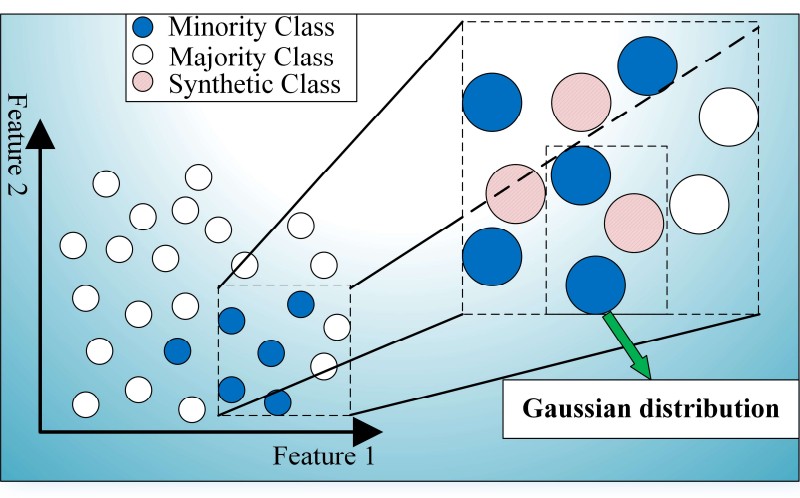

(**a**)

**Figure 5.** *Cont.*

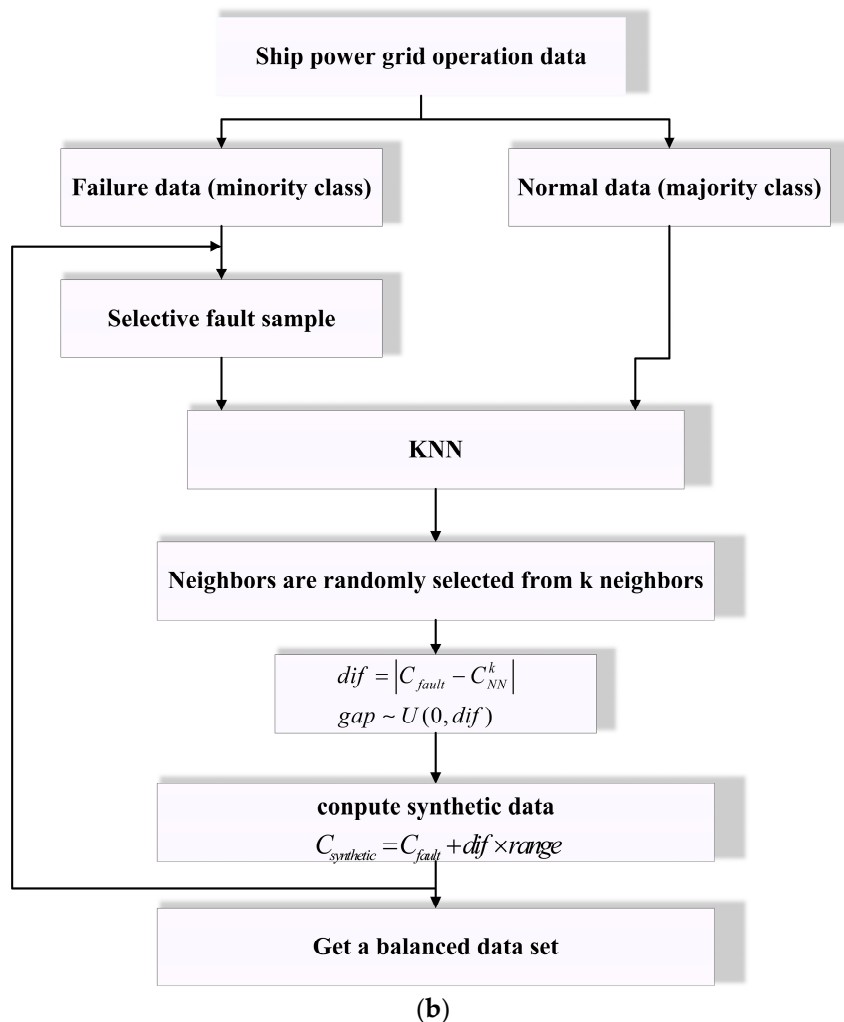

**(b)**

**Figure 5.** Based on Gaussian random variable SMOTE algorithm principle diagram and flowchart: (**a**) algorithm principle diagram; (**b**) algorithm flowchart.

### 3.3. Deep-Gated RNN-FCN

FCN, RNN, and LSTM are widely used neural networks for time series classification tasks [28,29]. Specifically, the data sequence of the ship's power grid represents a time series. To enhance the RNN's performance in capturing the dependency relationship of fault sequences and address the gradient vanishing issue, we replaced the computation of hidden states in the RNN with that in an LSTM. This technique creates a deep-gated RNN for our investigation. The modification of adopting the LSTM-based computation helps alleviate the problem of vanishing gradients in RNN while also improving its capacity to capture the dependence relation of fault sequences more effectively. We further constructed the Deep-Gated RNN-FCN model by integrating the deep-gated RNN with FCN. Through this combination, the deep-gated RNN enhances FCN's performance in time series classification, thereby improving the fault classification accuracy significantly.

RNN can be categorized into two types: those without hidden states and those with hidden states. In this study, we utilized the RNN with hidden states to capture and store contextual information from prior time steps. Given a regular RNN, the computation of the hidden variable $H_t$ at a time step $t$ is represented by Equation (17). This calculation is recurrent and directly depends on both the input at the current time step and the hidden variable from the preceding time step.

$$H_t = \phi(X_t W_{xh} + H_{t-1} W_{hh} + b_h) \tag{17}$$

where $H_t \in R^{n \times h}$ is the hidden variable, also known as the hidden state, at the current time step $t$. $H_{t-1}$ is the hidden variable at the previous time step $t-1$. $X_t \in R^{n \times d}$ represents the mini-batch input at the current time step. $W_{xh} \in R^{d \times h}$ and $W_{hh} \in R^{h \times h}$ are the weight parameters of the hidden layer, and $b_h \in R^{1 \times h}$ is the bias parameter of the hidden layer.

The calculation of the output $O_t$ at the time step $t$ is given by Equation (18):

$$O_t = H_t W_{hq} + b_q \tag{18}$$

In the equation, $W_{hq} \in R^{h \times q}$ represents the weight parameters of the output layer, and $b_q \in R^{1 \times q}$ represents the bias parameters of the hidden layer. In an RNN, the model parameters remain unchanged across different time steps, and the calculation logic is illustrated in Figure 6.

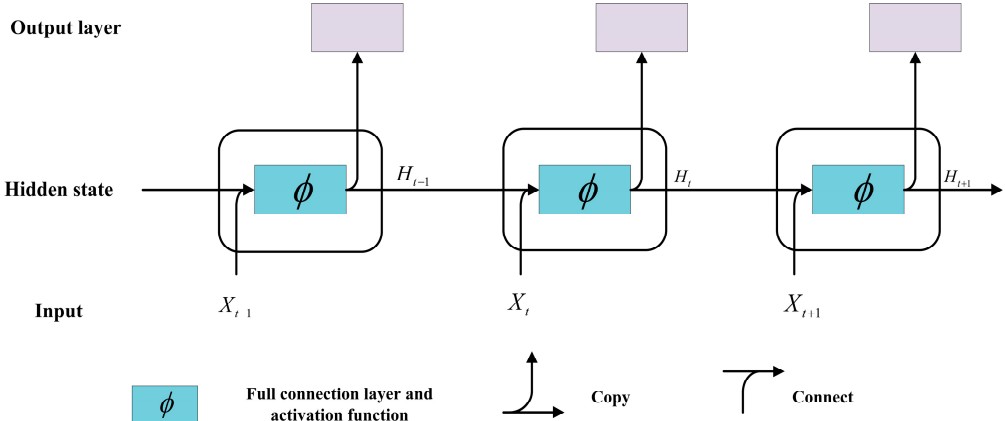

**Figure 6.** The computational logic diagram of a traditional RNN with hidden states.

LSTM networks are equipped with input gates, forget gates, output gates, and a dedicated hidden state called the memory cell. The computation of LSTM's hidden state is depicted in Figure 7, where the output gate manages the flow of information from the memory cell to the hidden state:

$$H_t = O_t \odot \tanh(C_1) \tag{19}$$

In the equation, $H_t \in R^{n \times h}$ represents the hidden state at the time step $t$, $O_t$ represents the output state, and tanh represents a function. It is ensured that the hidden state is within the range $[-1, 1]$.

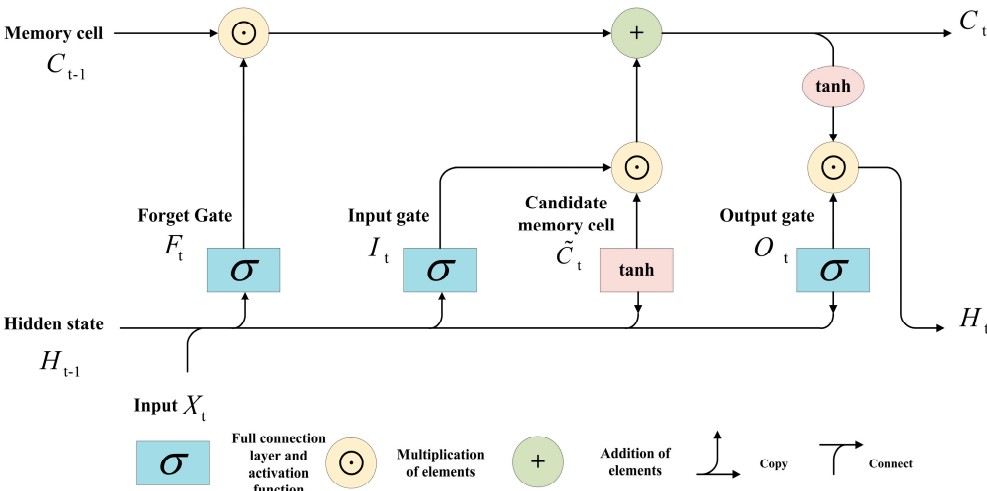

**Figure 7.** The computation logic diagram for the hidden state in an LSTM.

To replace the calculation of the hidden state in a traditional RNN with the hidden state calculation in an LSTM, we created a deep-gated RNN.

The FCN block, consisting of three stacked temporal convolutional blocks, with convolution kernel sizes of 128, 256, and 128, respectively, can be enhanced by incorporating the Deep-Gated RNN block. The input to the temporal convolutional blocks is the fault sequence signal. Let $X_t \in R^{F_0}$ be the input feature vector of length $F_0$ at the time step $t$, where $t \in (0, T]$. Assuming each layer has a time step of $T_l$ and there are $C$ classes, $y_t \in \{1, \ldots, C\}$ represents the true action label for each frame. The number of convolutional layers is $L$, and the convolutional kernels (or filters) of each layer are parameterized using Equation (20):

$$W^l \in R^{F_l \times d \times F_{l-1}}, b^l \in R^{F_l} \tag{20}$$

In the equation, $l$ represents the index of a certain layer, satisfying $l \in \{1, \ldots, L\}$, and $d$ represents the filtering time. On the $l$-th layer, $\hat{E}_t^l \in R^{F_l}$ is the i-th activated component, which is not normalized; $E^{l-1} \in R^{F_{l-1} \times T_{l-1}}$ is the activated matrix passed from the previous layer, which is normalized. As shown in Equation (21), $\hat{E}_t^l$ is a function of $E^{l-1}$. $f(\bullet)$ represents the rectified linear unit for any time step $t$:

$$\hat{E}_{i,t}^l = f\left(b_i^l + \sum_{t'=1}^{d} \langle W_{i,t',\cdot}^l, E_{\cdot,t+d-t'}^{l-1} \rangle\right) \tag{21}$$

Figure 8 illustrates the framework of the Deep-Gated RNN-FCN algorithm. The stacked time convolution blocks in FCN serve as the fault feature extraction module, and global average pooling is applied after the last convolution block to reduce the number of parameters in the model. Each time convolution block consists of a time convolution layer that performs BN on the input of each batch, accelerating the training of the neural network and improving model convergence. The activation function used is ReLU. The preprocessed power grid fault sequence passes via a dimension-shifting layer, which ultimately enables passing to the Deep-Gated RNN submodule. Dropout regularization is performed on each submodule to prevent overfitting and enhance the model's generalization capability, hence significantly reducing generalization errors. The global pooling layer and the Deep-Gated RNN's output are then concatenated and processed by the softmax fault classification layer. Thus, forming the network architecture of the Deep-Gated RNN-FCN algorithm. The input of the network is the preprocessed power grid fault dataset, and the output is the fault classification result.

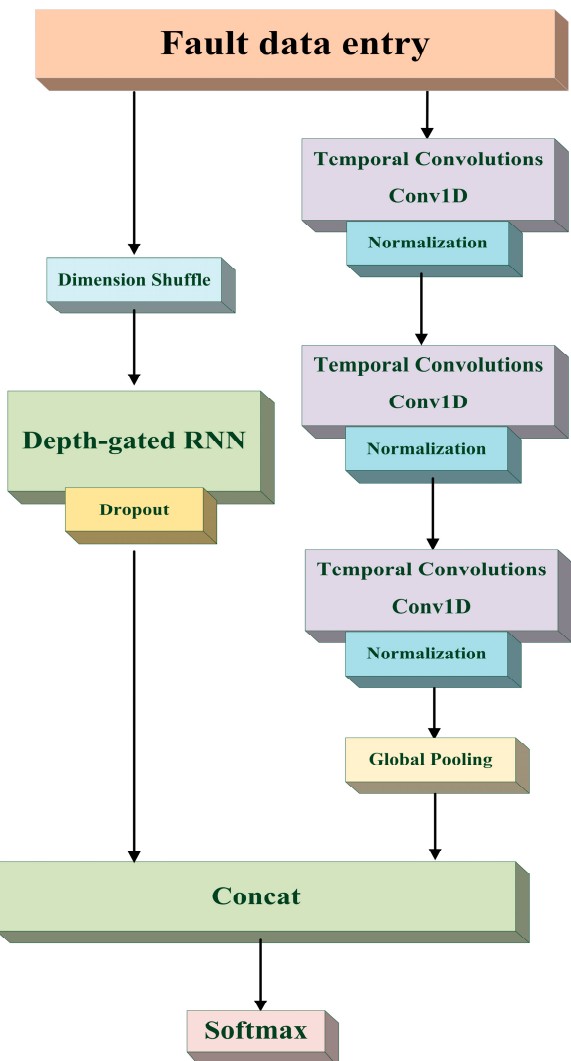

**Figure 8.** The framework diagram of the Deep-Gated RNN-FCN algorithm.

## 4. Analysis of Experimental Results and Comparison of Algorithms

The ship's integrated power system under study in this paper is the next-generation all-electric MVDC ship integrated power system proposed by the United States [30]. The ship has a reference mass of $1.429 \times 10^7$ kg and a rated voltage of 5 kV. In this paper, a simulation model of the system is established to validate the effectiveness of the proposed methods. The simulation includes the short-circuit fault at the generator side of the ship's MVDC power propulsion system and its fault diagnosis. A total of 11 types of fault electrical signals were collected, with 100,000 training data samples for each category. During the experimental process, the FCN block remained unchanged. Through hyperparameter search, the dropout rate was set to 80%, the initial learning rate was 0.001, and the learning rate adjustment strategy was as follows: after every 100 training iterations, if the model's validation score did not improve, the learning rate would be reduced by a factor of 0.794 until reaching the final learning rate of 0.0001. The model was trained using the Adam optimizer.

### 4.1. Model Evaluation Criteria

The proposed model will be evaluated based on accuracy, precision, ROC curve, and AUC. These evaluation criteria rely on a confusion matrix, which is illustrated in Table 4. The confusion matrix depicts the actual and predicted classes in its rows and columns, respectively. TN denotes the number of true negatives (negative instances that are

correctly classified), FP represents the number of false positives (negative instances that are incorrectly classified), FN indicates the number of false negatives (positive instances that are wrongly classified), and TP denotes the number of true positives (positive instances that are correctly classified).

**Table 4.** Confusion matrix.

| Actual | Predicted | Predicted Negative | Predicted Positive |
|---|---|---|---|
| Actual Negative | | TN | FP |
| Actual Positive | | FN | TP |

The accuracy (Acc) is defined as $(TP + TN)/(TP + FP + TN + FN)$ and represents the proportion of correctly classified instances among all the predicted samples.

Precision (Pre) is calculated as $TP/(TP + FP)$ and indicates the proportion of true positive instances among all predicted positive instances.

The ROC curve, which has $FPR = FP/(TN + FP)$ as its horizontal axis, is also called specificity and shows the proportion of correctly classified negative instances among all negative samples. The vertical axis represents $TPR = TP/(TP + FN)$, which is also known as recall or sensitivity and measures the model's ability to correctly identify positive instances. This curve evaluates the True Positive Rate and False Positive Rate at different thresholds. The AUC is a metric used to measure the area under the ROC curve and assess the performance of a binary classification problem. A higher AUC value, closer to 1, indicates better performance of the model, with a balance of high True Positive Rate and low False Positive Rate. A value closer to 0.5 indicates poor model performance.

*4.2. Simulation Results and Analysis*

The dataset was partitioned with an 80–20 ratio for training and testing, respectively. Figures 9 and 10 show the confusion matrices for the corresponding sets. The training set had a final accuracy of 98.8636%, while the testing set had an accuracy of 96.8182%.

Figure 11 shows the best accuracy and precision curves of the proposed fault diagnosis method in this paper after 100 iterations. The final accuracy of the test set was 97.35%, demonstrating the satisfactory performance of the proposed model based on MWDN and deep-gated RNN-FCN in terms of accuracy and precision.

The performance of the proposed model in this paper is evaluated using the ROC curve and its AUC. The evaluation criteria for AUC are as follows:

(1) AUC = 1: Perfect classifier that can perfectly distinguish positive and negative instances without any misclassifications, although it is challenging to achieve in practice.

(2) 0.5 < AUC < 1: Better than random guessing, indicating that the model has some classification ability.

(3) AUC = 0.5: Random guessing, indicating that the model's classification ability is equivalent to random selection.

(4) 0 < AUC < 0.5: Worse than random guessing, indicating that the model's classification ability is inferior to random selection.

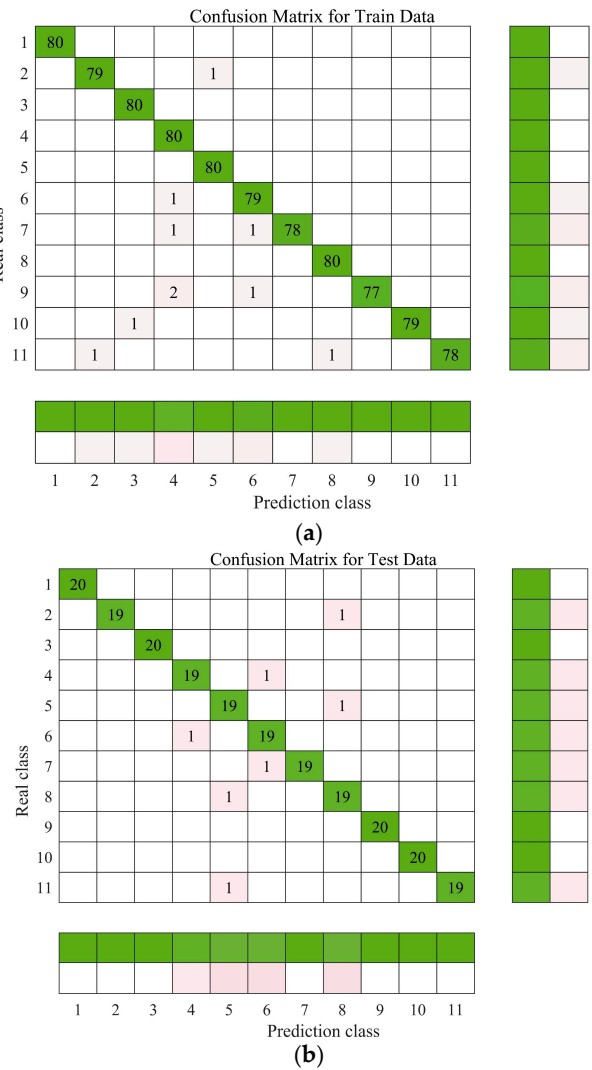

(**a**)

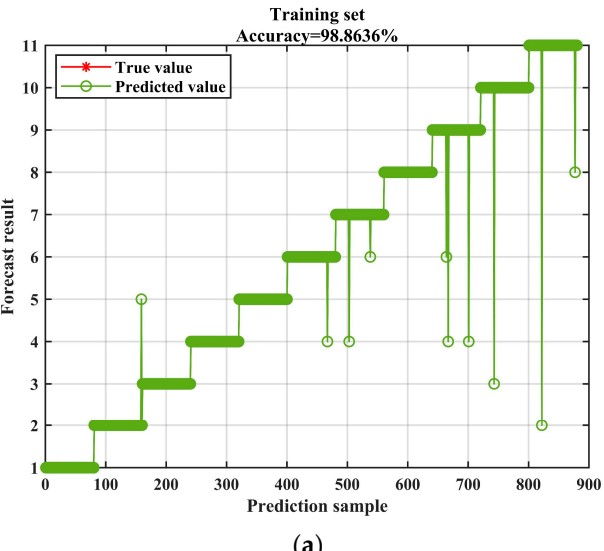

(**b**)

**Figure 9.** Confusion matrix diagram of the training set and test set: (**a**) confusion matrix for the training set and (**b**) confusion matrix for the testing set.

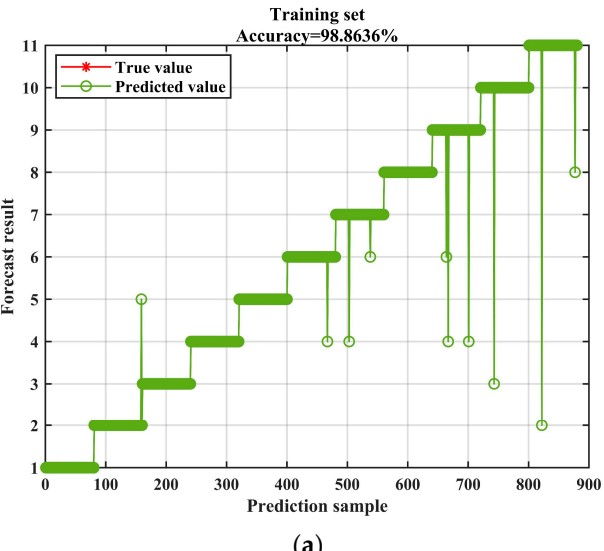

(**a**)

**Figure 10.** *Cont.*

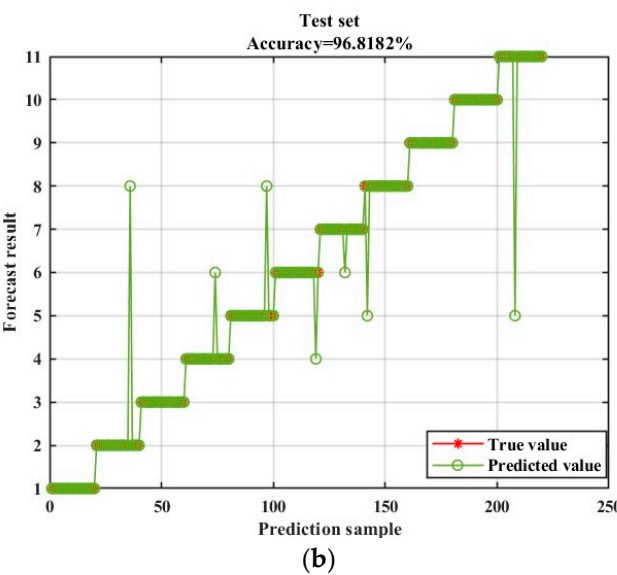

(**b**)

**Figure 10.** Training set and test set confusion matrix and result graph: (**a**) training set results graph and (**b**) test set results graph.

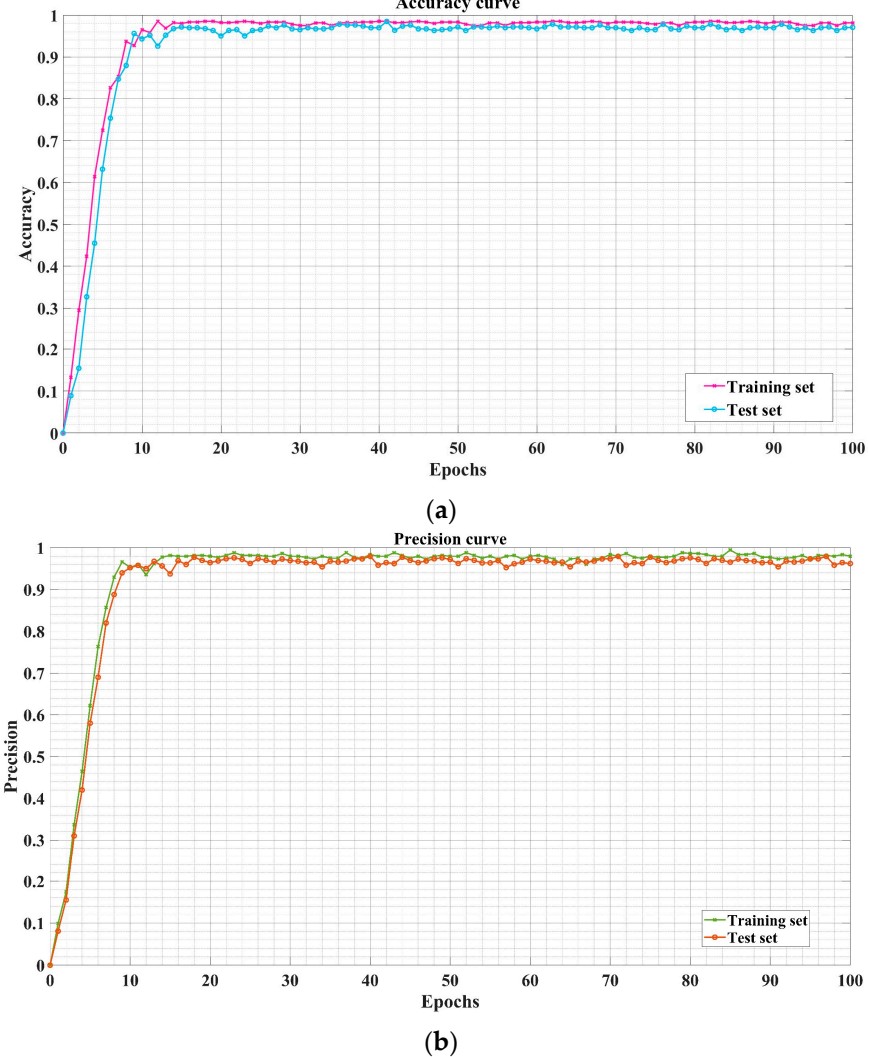

**Figure 11.** Best accuracy curve and accuracy curve: (**a**) accuracy and (**b**) precision.

The AUC value ranges from 0 to 1, with values closer to 1 indicating better model performance and values closer to 0.5 indicating poorer model performance. The ROC curve and AUC of the proposed algorithm in this paper are shown in Figure 12, with an AUC of 0.85. The curve $y = x$ serves as a comparison. The AUC indicates that the proposed method in this paper has good classification ability.

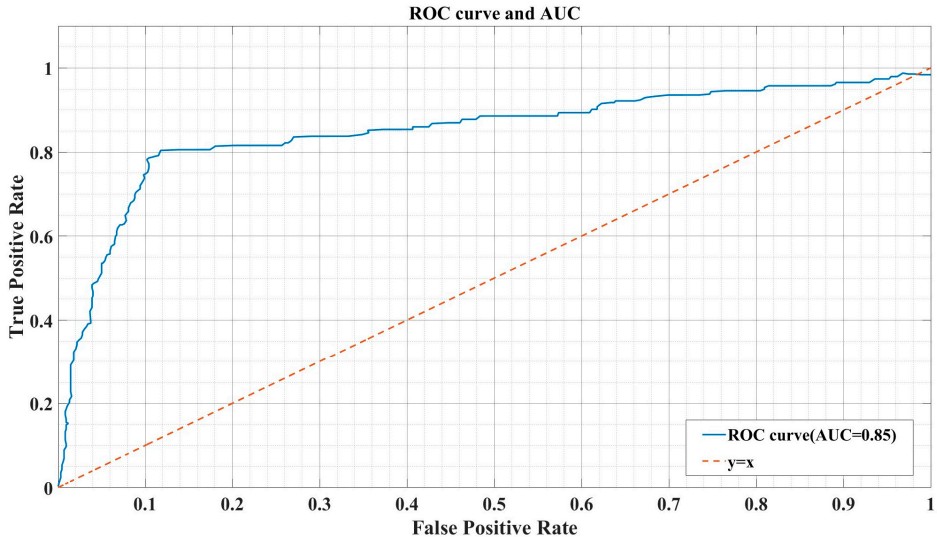

**Figure 12.** ROC curve and AUC of this algorithm.

### 4.3. Comparative Experimental Results with Other Algorithms

To evaluate the effectiveness of the diagnostic method presented in this research, we conducted a comparison with several other algorithms.

(1) Effectiveness of Deep-Gated RNN

In order to validate the effectiveness of extending FCN with Deep-Gated RNN, we created a standalone FCN model and applied the dataset for simulation. The outcomes in terms of accuracy, precision, and AUC of the test set are presented in Figure 13. The FCN model achieved an accuracy and precision of 92.53% and an AUC value of 0.76. From the graph, it is evident that the performance evaluation metrics for the Deep-Gated RNN-FCN surpass those of the standalone FCN model, indicating that integrating Deep-Gated RNN is an effective method to improve the performance of fault diagnosis using FCN.

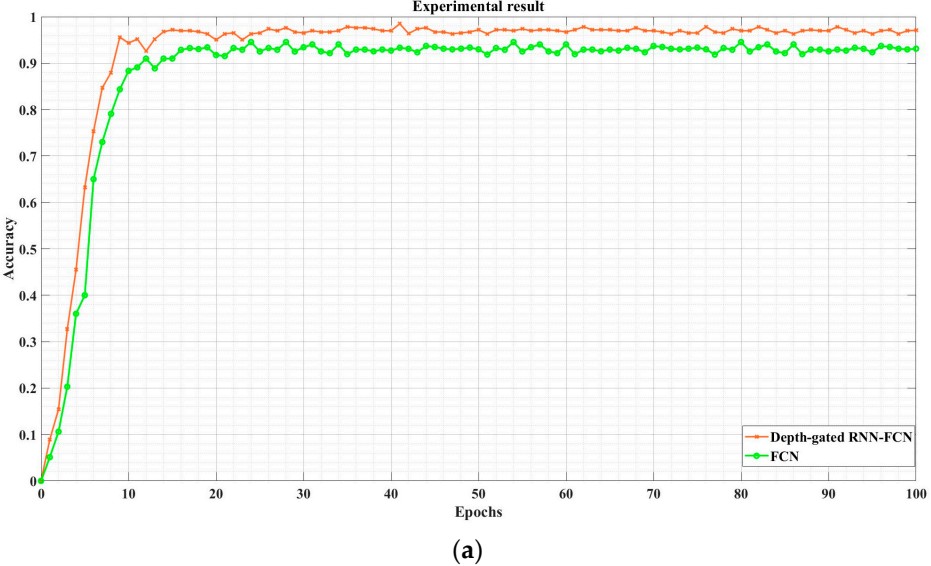

(**a**)

**Figure 13.** *Cont.*

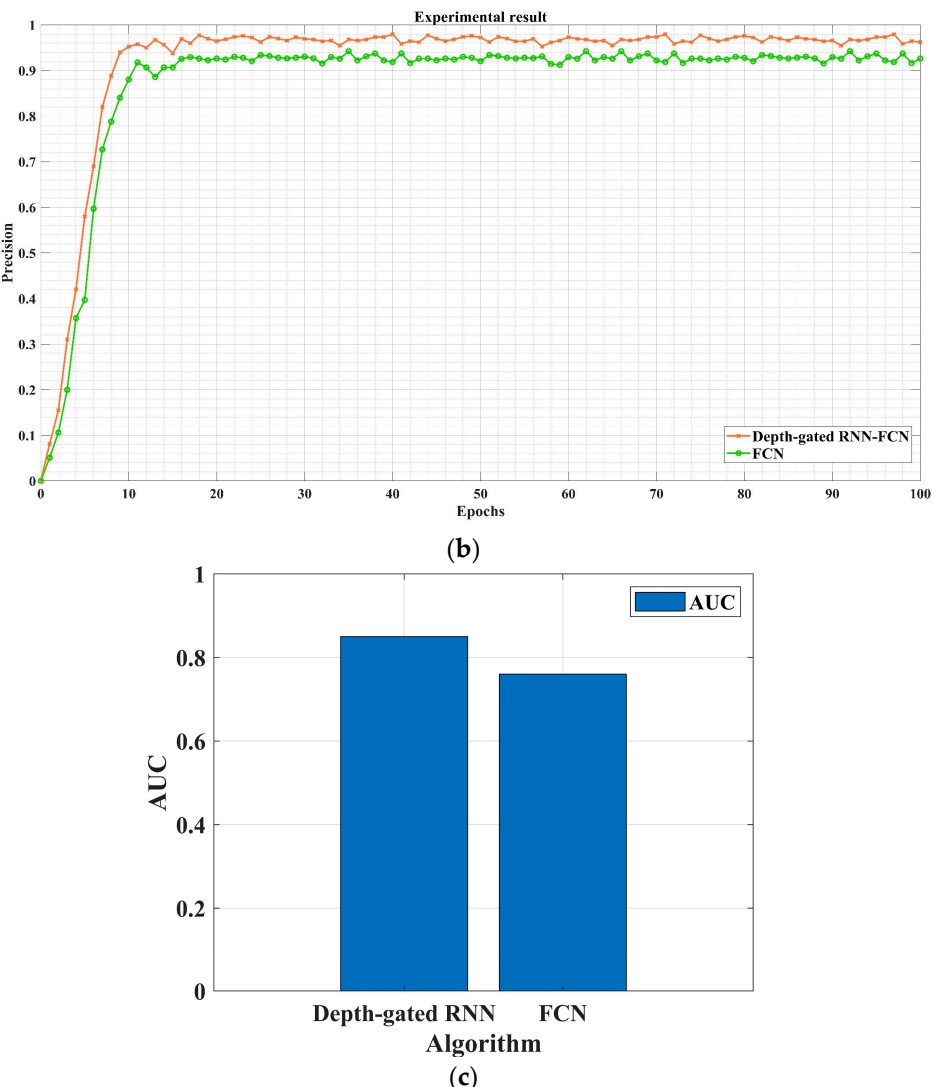

**Figure 13.** Comparison of the proposed algorithm with FCN: (**a**) accuracy, (**b**) precision, and (**c**) AUC.

(2) Effectiveness of Multi-level Wavelet Decomposition Network

In order to evaluate the effectiveness of MWDN in our proposed algorithm, we conducted a comparative simulation by excluding the MWDN component and contrasting the results with our algorithm. The simulation results are presented in Figure 14. The purpose of including the MWDN is to improve the accuracy of the final fault diagnosis outcomes. The comparative simulation revealed that the deep-gated RNN-FCN without MWDN achieved an accuracy of 91.26%, a precision of 92.43%, and an AUC value of 0.72. In contrast, our algorithm performed significantly better with an AUC value of 0.85. Based on the accuracy and precision, it is evident that the MWDN played a crucial role in enhancing the fault diagnosis performance of the deep-gated RNN-FCN. Hence, these findings confirm the effectiveness of MWDN.

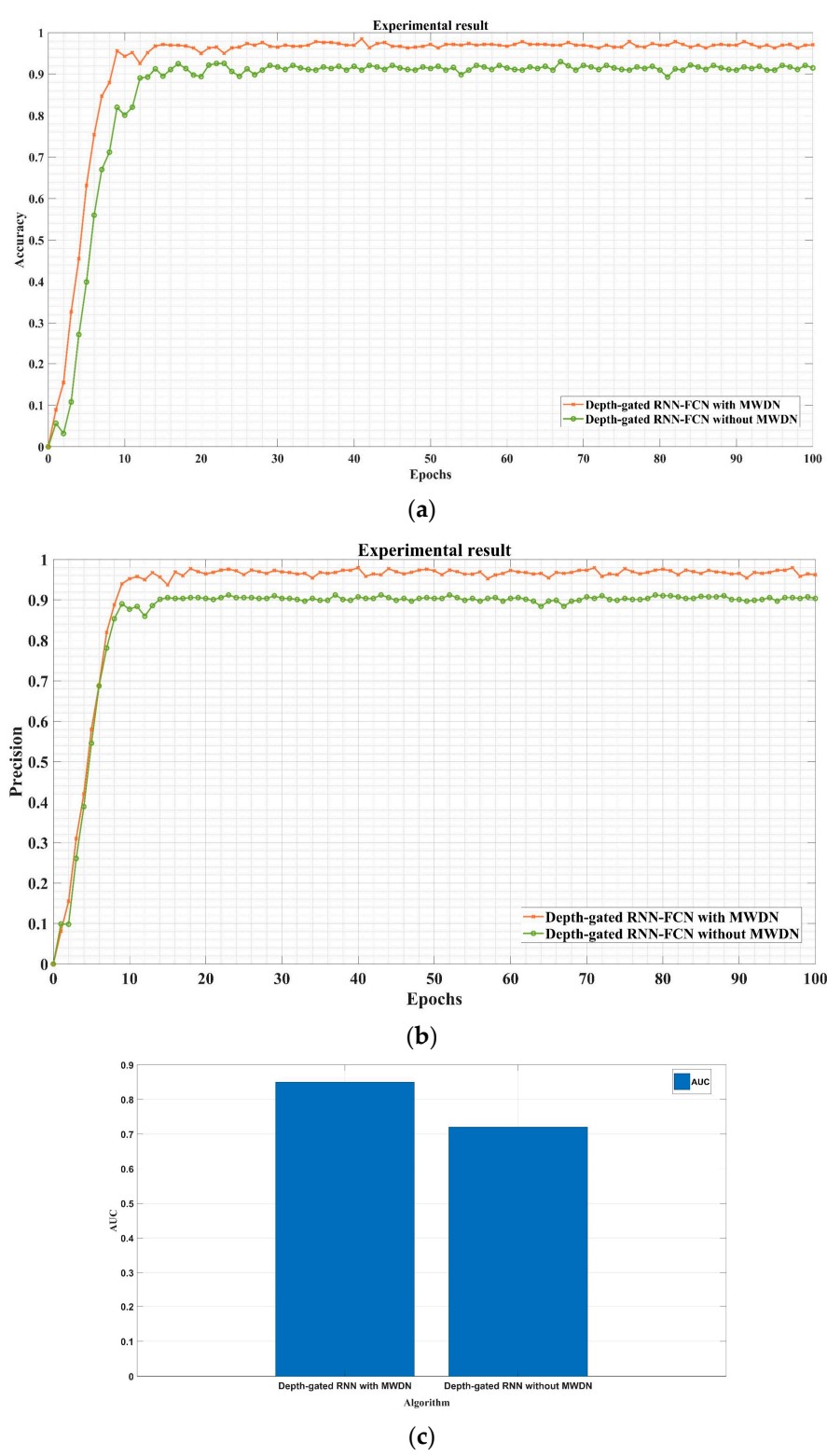

**Figure 14.** Comparison between the proposed algorithm and the algorithm without MWDN processing: (**a**) accuracy, (**b**) precision, and (**c**) AUC.

(3) The Effectiveness of SMOTE based on Gaussian Random Variable

The simulation results in Figure 15 compare the proposed Gaussian random variable-based SMOTE algorithm with the regular SMOTE algorithm and the algorithm that does not use SMOTE. The regular SMOTE algorithm achieved an accuracy of 91.64%, a precision of 91.89%, and an AUC value of 0.69. The algorithm without SMOTE yielded an accuracy

of 88.98%, a precision of 90.52%, and an AUC value of 0.60. In contrast, our proposed algorithm achieved significantly higher accuracy and precision than both the regular SMOTE and the algorithm without SMOTE. These findings illustrate the effectiveness of the contribution made by our Gaussian random variable-based SMOTE algorithm.

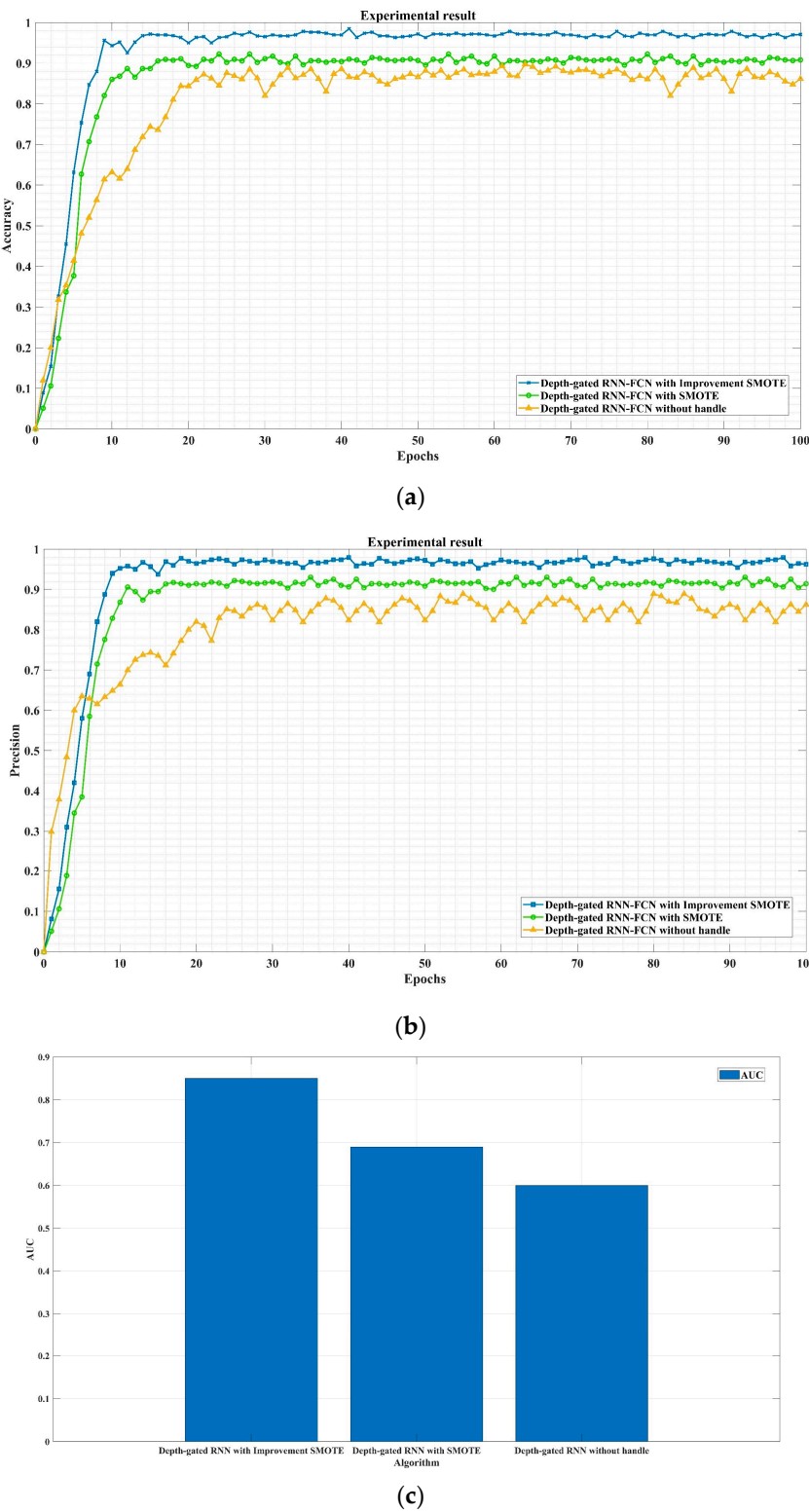

**(a)**

**(b)**

**(c)**

**Figure 15.** Comparison of improved SMOTE algorithm with original SMOTE algorithm and non-SMOTE algorithm: (**a**) accuracy, (**b**) precision, and (**c**) AUC.

(4) Comparison with other algorithms applicable to time series processing

As the power grid fault sequence constitutes a time series, it is appropriate to apply machine learning algorithms designed for time series feature extraction like LSTM-FCN, CNN-FCN, and TCN-FCN. Therefore, we compared the performance of our proposed algorithm with these three algorithms. Our simulation results, as illustrated in Figure 16, reveal that the accuracy of Deep-Gated RNN-FCN, LSTM-FCN, CNN-FCN, and TCN-FCN are 96.82%, 94.45%, 89.61%, and 90.00%, respectively. The precision is 97.35%, 93.42%, 90.60%, and 90.05%, respectively. The AUC values are 0.85, 0.80, 0.73, and 0.75, respectively. These metrics, taken together, demonstrate that our proposed algorithm outperforms the other three methods.

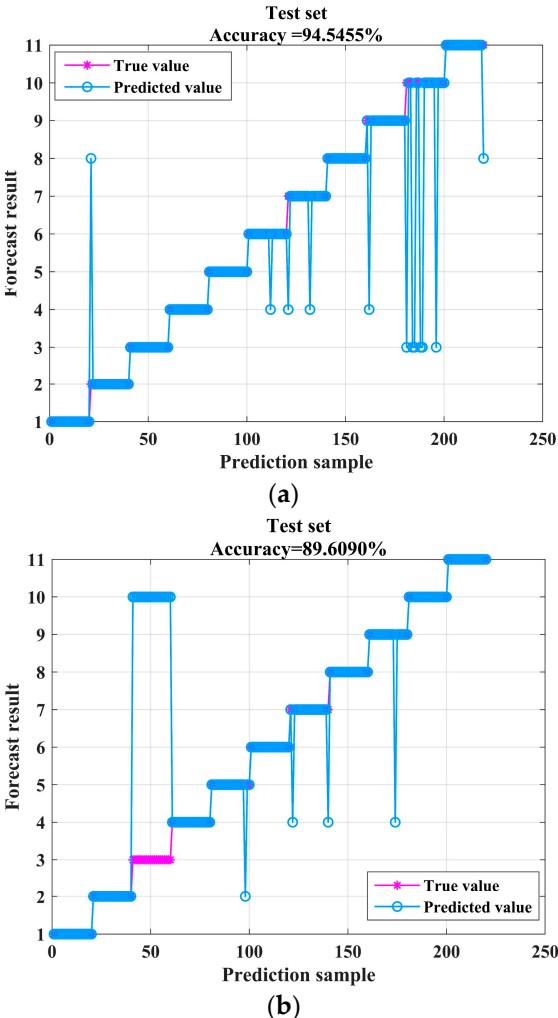

**Figure 16.** *Cont.*

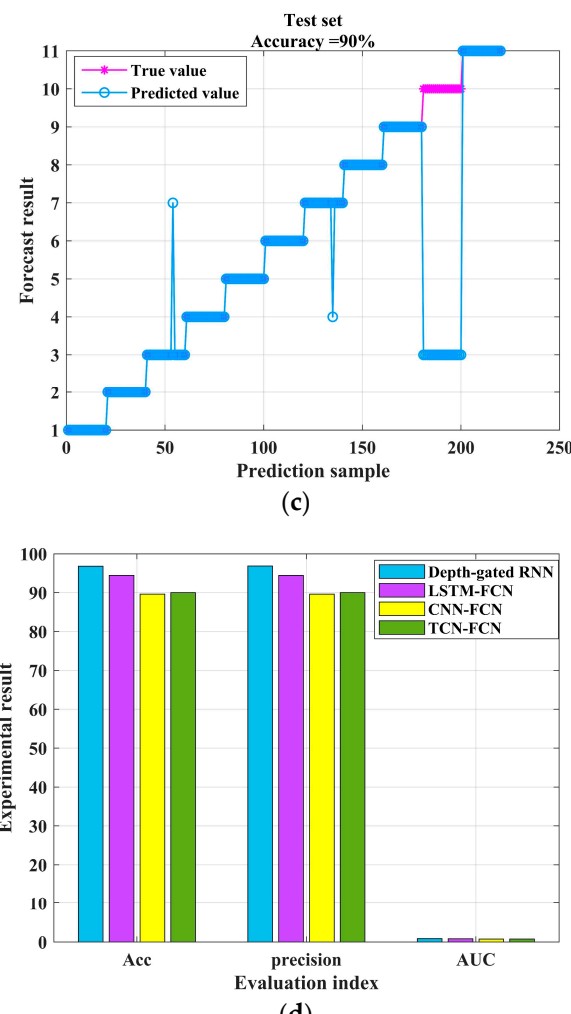

(c)

(d)

**Figure 16.** Comparison chart of the algorithm proposed in this paper with three other algorithms: (**a**) LSTM-FCN, (**b**) CNN-FCN, (**c**) TCN-FCN, and (**d**) summary of evaluation indicators for fault diagnosis algorithms.

## 5. Conclusions

The vessel's electrical propulsion system is responsible for powering the entire vessel, with the vessel's power system being centered around its three-phase synchronous generators. A critical short-circuit fault occurring at the generator's output terminal could instantaneously interrupt the power supply to the entire vessel, with severe negative implications for safety and vessel stability. Therefore, it is crucial to accurately diagnose and locate any short-circuit faults at the generator's terminal.

This study proposes a fault diagnosis method for short-circuit faults at the generator terminal based on MWDN and Deep-Gated RNN-FCN. The following research achievements have been obtained:

Firstly, MWDN-based feature extraction is proposed. MWDN decomposes raw signals into high-frequency sub-signals and low-frequency sub-signals, extracting time-domain and frequency-domain features of power grid signals while reducing noise interference and improving fault diagnosis accuracy.

Secondly, a Gaussian random variable-based SMOTE algorithm is proposed to address the class imbalance problem between normal and fault classes in power grid data. Compared to traditional SMOTE algorithms, the proposed approach reduces sample data synthesis generalization issues, making it more effective for handling power grid data.

Moreover, the study constructs a Deep-Gated RNN-FCN model. LSTM computation replaces the computation of hidden states in RNN, resulting in a Deep-Gated RNN. The Deep-Gated RNN model is then combined with the FCN model to extract features from input fault sequences. This approach can address the gradient vanishing issue and capture long-range dependencies in time series better, thus complementing the performance of FCN in time series classification. As a result, the performance of fault diagnosis improves significantly.

Finally, the softmax classifier is used for fault classification and results output. The proposed method is evaluated and tested via simulation, utilizing three performance indicators: accuracy, precision, and AUC. The accuracy rate is 96.82%; precision is 97.35%; and AUC is 0.85. This demonstrates the method's excellent performance. Comparative experiments with other algorithms show that the proposed method achieves greater accuracy and better classification of faults, validating its superiority and effectiveness. Furthermore, the proposed method exhibits good generalization ability, making it applicable in other fault diagnosis scenarios.

**Author Contributions:** Conceptualization, L.Z. and Z.Z.; methodology, Z.Z. and H.P.; software, L.Z.; validation, Z.Z. and H.P.; data curation, L.Z.; writing—original draft preparation, L.Z.; writing—review and editing, Z.Z. and H.P.; visualization, L.Z. All authors have read and agreed to the published version of the manuscript.

**Funding:** Supported in part by the State Key Laboratory of Smart Grid Protection and Operation Control under Grant SGNR0000KJJS2302140.

**Institutional Review Board Statement:** Not applicable.

**Informed Consent Statement:** Not applicable.

**Data Availability Statement:** Not applicable.

**Conflicts of Interest:** The authors declare no conflict of interest.

## Abbreviation

| Symbol | Implication |
|--------|-------------|
| MWDN | Multi-Level Wavelet Decomposition Network |
| MDWD | Multi-Level Discrete Wavelet Decomposition |
| RNN | Recurrent Neural Network |
| FCN | Fully Convolutional Network |
| SMOTE | Synthetic Minority Over-sampling Technique |
| MVDC | Medium Voltage Direct Current |
| WT | Wavelet Transform |
| EMD | Empirical Model Analysis |
| MSP | Morphological Signal Processing |
| PCA | Principal Component Analysis |
| ANN | Artificial Neural Network |
| SVM | Support Vector Machine |
| SDG | Signed Directed Graph |
| AI | Artificial Intelligence |
| CNN | Convolutional Neural Network |
| PSO | Particle Swarm Optimization |
| LSTM | Long Short-Term Memory |
| RF | Random Forest |
| FFT | Fast Fourier Transform |
| BN | Batch Normalization |
| AUC | Area Under Curve |
| $\varphi$ | the magnetic flux of each winding |
| $u$ | the voltage across each winding |
| $i$ | the current flowing through each winding |
| $r$ | the resistance of each winding |

| | |
|---|---|
| $X_d$ | the $d$-axis winding |
| $X_q$ | the $q$-axis winding |
| $X_f$ | the excitation winding |
| $X_D$ | the damping winding |
| $X_Q$ | the self-inductance |
| $X_{ad}$ | the mutual inductance between $d$-axis windings |
| $X_{aq}$ | the mutual inductance between $q$-axis windings |
| $E_{q0}''$ | the transient potential on the transverse axis before the short circuit |
| $E_{d0}''$ | the transient potential on the direct axis before the short circuit |
| $X_d''$ | the transient reactance on the direct axis |
| $X_q''$ | the transient reactance on the transverse axis |
| $E_{q0}'$ | the transient electromotive force before the short circuit |
| $X_d'$ | the transient reactance |
| $E_{q(0)}$ | the steady-state potential |
| $X_d$ | the steady-state reactance |
| $T_d''$ | the transient time constant on the $d$-axis |
| $T_d'$ | the transient time constant on the $q$-axis |
| $T_q''$ | the time constant for the decay of the DC component |
| $x_{n+k-1}^l(i)$ | the $n+k-1$-th element of the $i$-th level low-frequency subsequence |
| $x^l(0)$ | the input sequence |
| $b^h(i)$ $b^l(i)$ | trainable bias vectors, initialized with random values that are infinitesimally close to zero |
| $\sigma(\bullet)$ | the activation function |
| $C_{fault}$ | a fault data point |
| $C_{NN}^k$ | a randomly selected neighbor from $k$ nearest neighbors following a uniform probability distribution for fault class data |
| $P_{random}$ | a random value |

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
