# Peer review of "Diagnostic Method for Short Circuit Faults at the Generator End of Ship Power Systems Based on MWDN and Deep-Gated RNN-FCN"

_jmse, doi:10.3390/jmse11091806_

Round 1
Reviewer 1 Report
Comments and Suggestions for Authors
This paper discusses a diagnostic method for short-circuit faults based on intelligent approaches. The topic is actual nowadays. However, there are some comments to be considered:
1. The abstract is supposed to be informative and brief. Please try to avoid abbreviations in it.
2. More relevant and latest references must be added in order to describe the today’s situation in the field.
3. The text is a bit overloaded with mathematics. Try to use some flowchart to describe the methodology.
4. The quality of pictures must be improved.
5. There are several spelling and grammar mistakes in the text.
6. Please try to make graphs more informative.
7. What is the main novelty of this study?
8. What kind of signals are prioritized in short-circuit detection (voltage, current, vibration, etc)? Which of them shows in the best way the fault existence?
9. Why there was chosen a Wavelet for feature extraction over Fast Fourier transform?
10. Is this approach is also applicable to other fault types?
Comments on the Quality of English Language
Minor editing of English language is required.
Reviewer 2 Report
Comments and Suggestions for Authors
1) The Abstract is too general and only descriptive. In the Abstract the Authors should add some of the most important results obtained in this research (its exact values). Such addition will highlight the novelty of the presented paper already in the Abstract – at the moment from the abstract cannot be seen any exact novel elements which are obtained in this paper.
2) Due to the high number of abbreviations, symbols and markings used throughout the paper text, in the paper should be added a Nomenclature inside which will be listed and explained all mentioned elements in one place. I strongly believe that the Nomenclature will notably improve paper readability (it will be notably helpful to me as well as to many other readers). Also, it will be much understandable to explain the parameters of the equations in the Nomenclature, instead below equations.
3) The Authors should avoid division of the Tables on two pages (for example, Tables 1 and 3). Such presentation makes at least some parts of the table difficult to understand.
4) Figures should be enlarged in general. Many details cannot be clearly visible even if using high zoom (for example in Figures 3 and 4), many figures are blurry and require corrections (for example Figure 11 and Figures from 13 to 15). The details on the mentioned figures are highly important, but they are hardly visible or understandable at the moment. So, the corrections are required.
5) At the beginning of Section 4 the Authors have stated that: “A total of 11 types of fault electrical signals were collected”. There are no any details how the dataset is collected. Missing of any explanations leads me to a question – are the used dataset product of some kind of simulation model, or the data are collected during the ship exploitation?
If the data are collected during ship exploitation, then the details about the ship, measurement equipment, measuring procedure and overall data accuracy/precision should be presented and explained.
If the dataset is a product of a simulation model – than a validation should be presented (direct comparison of simulated data and data obtained during real exploitation conditions). Without proper and exact validation, there is no numerical model which results can be considered as sufficiently accurate and precise to be a proper representation of the data obtained in real exploitation conditions.
Anyhow, I believe that this is a highly important task which must be performed and presented in the paper – without exact and proper explanations how the data used in the analysis are obtained and can they be proper representative of the real exploitation conditions, this entire paper can be theoretical only, without any exact connection with real exploitation conditions. That (theoretical only) researches can be nice to see, but they are not rounded and complete scientific paper (at least in my opinion).
6) Throughout the paper text should be added exact calls on each Figure and Table. Please, avoid terms like “Table above” or “Figure below” – the exact calls should be used in the paper text.
7) As the Abstract, the Conclusions section should also be improved with the most important obtained results (its exact values). Also the Conclusions seem to be too descriptive and general, without any exact details obtained in the research.
8) I highly suggest enlarging the List of References with more recent literature from this research field.
Final remarks: This is an interesting paper. The Authors have developed a new algorithm which clearly outperforms other standard algorithms used for this kind of problems. During the revision process, a special attention the Authors should place on the explanations and details related to the used dataset and on improving and enlarging figures.
Comments on the Quality of English LanguageMinor editing of English language required.
Reviewer 3 Report
Comments and Suggestions for Authors
The authors propose a generator terminal short-circuit fault diagnosis method based on a hybrid model combining Multi-Level Wavelet Decomposition Network (MWDN), Deep Gated Recurrent Neural Network (RNN), and Fully Convolutional Network (FCN) .
Figure 1 should be revised by the authors (enlarge adequately the textbox).
One nomenclature table would be fine.
Additional references in hte main text should be added in the right place (e.g. SMOTE, MWDN, softmax).
Figure 3 should be more explained related to the already inserted scientific description.
Recent references (2023) should be added.
One screenshot of the implemented software file should be added.
Round 2
Reviewer 1 Report
Comments and Suggestions for Authors
Thank you for considering my suggestions. However, there ares still several comments:
1. The keyword must be revised and changed.
2. Several spelling and grammar mistakes can be found in the text.
3. Tables must be revised. What does *Etc* mean in the table?
4. Still, the quality of some pictures e.g. Fig. 8 must be improved.
Comments on the Quality of English LanguageMinor editing of English language is required.
Author Response
请参阅附件。

Reviewer 2 Report
Comments and Suggestions for Authors
The Authors have performed the most of proposed corrections/additions/improvements. However, the most important element mentioned in my previous review remains unsolved. The problematic is that missing this element makes the whole paper highly questionable and possibly wrong.
The unresolved element during the revision process is related to my previous comment 5.
The Authors have stated that used data are obtained by numerical simulation. I have nothing against that approach, but such data must be validated and verified by proper validation (direct comparison of the numerical simulation results and data measured during ship exploitation). The Authors can choose in which way the validation will be performed.
Without any validation (as it is at the moment in the paper), there is no any evidence that the data obtained by numerical simulation are sufficiently accurate and precise and that such data can be a proper representative of the real exploitation results. Using any kind of algorithms on a data which did not represent real exploitation conditions can be mathematically interesting, but scientifically worthless (at least in my opinion).
Finally, without any validation, this paper is at least highly questionable and cannot have my recommendation for publication.
Therefore, my opinion is that the paper requires another revision and performing proper validation which will confirm that used data are proper representative of the real exploitation conditions.
Reviewer 3 Report
Comments and Suggestions for Authors
The paper has been improved by the author according to the reviewer's remarks.
The authors should check carefully the text, including in figures (see the textbox in Fig.1- Harmonic wave).
